# Is the vIL-10 Protein from Cytomegalovirus Associated with the Potential Development of Acute Lymphoblastic Leukemia?

**DOI:** 10.3390/v17030435

**Published:** 2025-03-18

**Authors:** Ruvalcaba-Hernández Pamela, Mata-Rocha Minerva, Cruz-Muñoz Mario Ernesto, Mejía-Aranguré Juan Manuel, Sánchez-Escobar Norberto, Arenas-Huertero Francisco, Melchor-Doncel de la Torre Silvia, Rangel-López Angélica, Jiménez-Hernández Elva, Nuñez-Enriquez Juan Carlos, Ochoa Sara, Xicohtencatl-Cortes Juan, Cruz-Córdova Ariadnna, Figueroa-Arredondo Paula, Arellano-Galindo José

**Affiliations:** 1Laboratorio de Virología, Unidad de Investigación en Enfermedades Infecciosas, Hospital Infantil de México Federico Gómez, Mexico City 06720, Mexico; pame.maeunam@gmail.com (R.-H.P.); melchor.sp@gmail.com (M.-D.d.l.T.S.);; 2Posgrado en Ciencias Biológicas, Universidad Nacional Autónoma de México, Mexico City 04510, Mexico; 3Unidad de Investigación en Genética Humana, Hospital de Pediatría, Centro Médico Nacional Siglo XXI, Instituto Mexicano del Seguro Social, Avenida Cuauhtémoc 330, Doctores, Ciudad de México 06720, Mexico; mmata@conacyt.mx (M.-R.M.); venancio.saes@gmail.com (S.-E.N.); 4Facultad de Medicina, Universidad Autónoma del Estado de Morelos, Cuernavaca 62209, Mexico; mario.cruz@uaem.mx; 5Laboratorio de Genómica Funcional del Cáncer, Instituto Nacional de Medicina Genómica (INMEGEN), Mexico City 14610, Mexico; jmejia@inmegen.gob.mx; 6Facultad de Medicina, Universidad Nacional Autónoma de México, Mexico City 04510, Mexico; 7Facultad de Medicina y Cirugía, Universidad Autónoma “Benito Juárez” de Oaxaca, Oaxaca City 68120, Mexico; 8Laboratorio de Investigación en Patología Experimental, Hospital Infantil de México Federico Gómez, Ciudad de México 06720, Mexico; farenashuertero@yahoo.com.mx; 9Departamento de Oncología, Hospital Pediátrico Moctezuma SEDESA, Universidad Autónoma Metropolitana, Mexico City 09769, Mexico; elvajimenez@yahoo.com; 10Unidad de Investigación Médica en Epidemiología Clínica, Hospital de Pediatría, Centro Médico Nacional Siglo XXI, Instituto Mexicano del Seguro Social, Mexico City 06720, Mexico; jcarlos_nu@hotmail.com; 11Laboratorio de Bacteriología Intestinal, Hospital Infantil de México Federico Gómez, Mexico City 06720, Mexico; saraariadnah@hotmail.com (O.S.); juanxico@yahoo.com (X.-C.J.); 12Laboratorio de Inmunoquímica, Hospital Infantil de México Federico Gómez, Mexico City 06720, Mexico; ariadnnacruz@yahoo.com.mx; 13Escuela Superior de Medicina del Instituto Politécnico Nacional, Mexico City 11340, Mexico; 14Centro Interdisciplinario de Ciencias de la Salud Unidad Milpa Alta Instituto Politécnico Nacional, Mexico City 12000, Mexico

**Keywords:** vIL10, ALL (Acute lymphoblastic leukemia), hCMV (Cytomegalovirus), latent proteins

## Abstract

Leukemia is a hematologic malignancy; acute lymphoblastic leukemia (ALL) is the most prevalent subtype among children rather than in adults. *Orthoherpesviridae* family members produce proteins during latent infection phases that may contribute to cancer development. One such protein, viral interleukin-10 (vIL-10), closely resembles human interleukin-10 (IL-10) in structure. Research has explored the involvement of human cytomegalovirus (hCMV) in the pathogenesis of ALL. However, the limited characterization of its latent-phase proteins restricts a full understanding of the relationship between hCMV infection and leukemia progression. Studies have shown that hCMV induces an inflammatory response during infection, marked by the release of cytokines and chemokines. Inflammation may, therefore, play a role in how hCMV contributes to oncogenesis in pediatric ALL, possibly mediated by latent viral proteins. The classification of a virus as oncogenic is based on its alignment with cancer’s established hallmarks. Viruses can manipulate host cellular mechanisms, causing dysregulated cell proliferation, evasion of apoptosis, and genomic instability. These processes lead to mutations, chromosomal abnormalities, and chronic inflammation, all of which are vital for carcinogenesis. This study aims to investigate the role of vIL-10 during the latent phase of hCMV as a potential factor in leukemia development.

## 1. Introduction

Leukemia is a malignant condition characterized by the abnormal proliferation of specific lineages of hematopoietic progenitors in blood cells [1]. This disease has become increasingly relevant because of its rising incidence in recent years. According to reports, leukemia was the 13th most diagnosed cancer and the 10th leading cause of cancer-related deaths globally, with approximately 487,000 new cases and 305,000 fatalities [2]. Acute lymphoblastic leukemia (ALL) is the most prevalent form of leukemia in children, with an incidence rate of 3.3 cases per 100,000 children. In adults, the incidence rate is lower, at approximately 1.6 cases per 100,000 individuals [3,4]. The pathophysiology of leukemia is recognized as multifactorial, and the role of infections in leukemogenesis should not be underestimated. While evidence linking human cytomegalovirus (hCMV) infection to an increased risk of leukemia is still limited, some studies have suggested that “in utero” exposure to hCMV may alter neonatal cytokine levels [5]. Furthermore, the expression of viral IL-10 by hCMV could modulate the immune response [6,7]. Research has indicated that the natural history of pediatric leukemia follows a two-hit model. The first “hit” occurs “in utero”, arising from fetal stress, wherein certain genetic alterations may be tolerated. The second hit is typically initiated by an external factor, such as a delayed infection or an absence of infections, which can disrupt the immune system. This disruption may lead to proliferative stress in the bone marrow, ultimately resulting in secondary mutations that are crucial for the development of leukemia [8].

## 2. hCMV and Leukemogenesis

### 2.1. Human Cytomegalovirus

hCMV, also referred to as human herpesvirus 5 (HHV-5), is a member of the *Orthoherpesviridae* family and the β-Herpesvirinae subfamily. It features a linear double-stranded DNA genome that spans approximately 240 kilobases, and the virus itself measures between 150 and 200 nanometers, making it one of the largest viruses in its family. The viral genome is encapsulated in an icosahedral capsid made up of around 162 capsomers [9,10], which is surrounded by a tegument and a lipid envelope. The surface glycoproteins of the virus facilitate the fusion of viral and cellular membranes, allowing for viral entry. Additionally, the tegument proteins play a vital role in transporting the viral genome to the cell nucleus following entry (Figure 1) [9,11].

### 2.2. Epidemiology

hCMV infection is one of the most prevalent viral infections globally, affecting approximately 60% of the adult population in developed countries and around 90% in developing nations [12]. In healthy individuals, hCMV is typically asymptomatic, as the antiviral immune response effectively controls the infection. However, it can lead to severe disease in immunosuppressed patients, such as those with AIDS or organ transplant recipients, causing significant complications including encephalitis, pneumonia, pericarditis, myocarditis, splenitis, and various other conditions, all of which carry risks of increased morbidity and mortality [12,13,14].

Transmission of hCMV commonly occurs through body fluids, including urine, saliva, tears, semen, and breast milk. The incubation period for the virus ranges from 3 to 12 weeks. Additionally, hCMV can be transmitted vertically from mother to fetus or newborns via breast milk, resulting in congenital defects such as deafness, vision loss, learning delays, and neurological impairments. This transmission happens when the virus crosses the placenta, leading to fetal infection, brain damage, and abnormal development during pregnancy. Such defects may arise from either a primary maternal infection or viral reactivation. Furthermore, the virus can be acquired during birth or breastfeeding, which may result in symptoms manifesting later in the postnatal period, complicating timely diagnosis and potentially leading to irreversible damage [10,15,16,17,18].

### 2.3. Role of Immune Evasion of hCMV in Tumorigenesis

Viruses have coevolved with their hosts, developing sophisticated strategies to evade immune responses and manipulate cellular pathways for replication and dissemination [19,20]. A variety of immune evasion mechanisms enable viruses to survive and regulate their biological cycles by influencing several cellular functions, including differentiation, proliferation, growth, glucose metabolism, and apoptosis. The Table 1 shows some of the mechanisms of immune evasion known to hCMV.

This enables viruses to commandeer the host cell’s machinery to produce their proteins and sustain an active infection [36,37]. Like other herpesviruses, hCMV establishes a lifelong infection following the initial infection, entering a state of latency mainly in cells of the myeloid lineage. Under certain conditions, the virus can reactivate, leading to the production of new viral particles. The specialized mechanisms of hCMV have evolved through dynamic interactions with its host. Viral proteomics serves as a vital tool that plays various regulated roles in the viral infectious cycle [19,20,38]. Could the manipulation of cellular processes be associated with the expression of hCMV proteins during different phases of its biological cycle, possibly leading to cellular transformation? Numerous studies have demonstrated that proteins and microRNAs expressed by various viruses can alter cellular pathways. These modifications may enable viruses to evade the immune response, ultimately affecting cellular transformation. Such phenomena have been observed with human papillomavirus, Epstein–Barr virus, and hepatitis B virus [39]. hCMV has developed multiple mechanisms of immune evasion, allowing it to coexist with its host. This interaction influences various immune processes and alters cell signaling pathways, thereby promoting the progression and spread of the previously defined tumor as on modulation [40,41].

### 2.4. hCMV Viral Life Cycle and the Latency Proteins

The life cycle of hCMV comprises two distinct phases: lytic and latent. The lytic phase is marked by the active production of viral particles, enabling the infection of new cells. In contrast, the latent phase is nonproductive and contributes to immunomodulation functions [38,42]. The virus encodes at least 200 genetic products that are crucial for producing viral proteins, which are essential for assembling new particles and enhancing evasion of the host immune response [11,38,42,43]. During the lytic phase, specific genes are expressed at various times to produce proteins with specialized functions. The immediate-early proteins (IE) are generated within the first 0 to 4 h postinfection (p.i.), playing a vital role in regulatory processes during viral transcription. They are also key players in immune evasion and dysregulation, which aid the virus in persisting within the host. The delayed-early proteins, produced between 4 to 48 h p.i., actively contribute to DNA replication and transcriptional regulation [38,44]. Much less is known about the proteins expressed during the latent phase in comparison with those expressed during the lytic phase. However, these proteins seem to be involved in viral assembly, given their role in coding for structural proteins necessary for virion assembly and cellular exit. Additionally, they may significantly influence cell surface protein expression and disrupt cell adhesion, immune responses, metabolism, transport, and programmed cell death [38,44,45,46]. These proteins could function as critical regulators of immune evasion, inhibiting antigen presentation through various mechanisms that disturb the host cytokine network. Key examples include LUNA (latency-associated unidentified nuclear antigen), US28, and vIL-10, which encodes the viral IL-10 protein, a homolog of the human IL-10 cytokine (Figure 2) [47,48,49].

### 2.5. Viral Infection and Malignance

#### 2.5.1. Oncogenic Viruses

As obligate intracellular parasites, viruses rely on the host’s cellular machinery for replication, strategically co-opting it to facilitate their own replication cycle. This interaction establishes a complex molecular conflict in which the virus endeavors to propagate and ensure its persistence, while the host cell activates intrinsic defense mechanisms to counteract infection. Within this ongoing host–pathogen arms race, certain viruses express oncogenic proteins capable of inducing cellular transformation, thereby contributing to tumorigenesis [50]. The mechanisms through which the most prevalent oncogenic viruses contribute to carcinogenesis are as follows.

(a) Epstein–Barr virus. Epstein–Barr virus (EBV) was the first oncogenic virus to be identified, and its role in tumorigenesis is mediated by multiple molecular mechanisms. Key viral macromolecules implicated in this process include the three latent membrane proteins (LMP1, LMP2A, and LMP2B), along with the Epstein–Barr nuclear antigens (EBNAs). Specifically, the EBNA family comprises five nuclear antigens: EBNA1, EBNA2, EBNA3A, EBNA3B, and EBNA3C. Moreover, EBV-encoded microRNAs play a pivotal role in modulating epigenetic regulatory networks and signaling pathways that drive tumor initiation and progression [51].

(b) Papillomavirus. Human papillomavirus (HPV) has been implicated in the development of various cancers, including those of the genital, head and neck, oral, and anal regions, and may also contribute to the pathogenesis of other cancer types that are not yet fully elucidated [52]. The viral oncoproteins E6 and E7 play a pivotal role in promoting genetic instability. E6 exerts its effects by interacting with the tumor suppressor protein p53, leading to its proteasomal degradation, and may also bind to additional regulatory proteins, such as Bak, which is involved in the apoptotic pathway. In contrast, E7 interacts with the retinoblastoma protein (pRB), thereby facilitating the deregulation of cell cycle progression and promoting uncontrolled cellular proliferation [53].

(c) Hepatitis viruses. Chronic infection with hepatitis B (HBV) and hepatitis C (HCV) represents major etiological factors in the pathogenesis of hepatocellular carcinoma. HBV, a hepadnavirus, has the capacity to induce epigenetic modifications and interfere with DNA repair mechanisms, as well as to perturb key cellular signaling pathways. At the molecular level, HBV persists as covalently closed circular DNA (cccDNA), which maintains an episomal form during long-term infection. This cccDNA serves as a template for the transcription of multiple RNA species, one of which encodes the HBx protein. HBx is integral to hepatocarcinogenesis, primarily through its contribution to the dysregulation of cell cycle progression and disruption of cellular homeostasis [54]. Additionally, the integration of viral DNA into the host genome can induce genomic instability, thereby contributing to the onset of carcinogenesis. In contrast, hepatitis C virus (HCV), a member of the *Flaviviridae* family, encodes a nonstructural protein termed NS3. This protein exerts its effects primarily by interacting with the tumor suppressor protein p53, thereby inhibiting its functional activity, while also modulating other critical proteins involved in the regulation of cellular signaling networks [55].

(d) Human T-lymphotropic virus 1. HTLV-1 is a retrovirus that specifically targets T cells and is identified as the etiological agent of adult T-cell leukemia/lymphoma. The virus encodes several regulatory proteins, including *Tax*, *Rex*, *p21*, *p12*, *p30*, and HBZ (HTLV-1 basic leucine zipper factor), which play pivotal roles in the oncogenic process. These proteins are instrumental in viral persistence, genomic stability, and latency. Among them, *Tax* is particularly critical, as it mediates the dysregulation of key genes involved in the regulation of the cell cycle, thereby facilitating uncontrolled cellular proliferation [4,56].

(e) Merkel cell poliomavirus. Merkel cell polyomavirus (MCPyV), a member of the Polyomaviridae family, has been implicated in the etiology of Merkel cell carcinoma. Evidence suggests that this small virus encodes a conserved region (CR-1) within its N-terminal domain, which exhibits significant homology to the E1A protein of adenoviruses. This region demonstrates oncogenic potential, facilitating the binding to the retinoblastoma (RB) tumor suppressor, thereby contributing to the molecular pathways involved in tumorigenesis [57].

#### 2.5.2. A Comparison Between hCMV and Other Oncogenic Viruses

As previously shown, viruses with established oncogenic potential employ a range of mechanisms throughout their lifecycle to disrupt the homeostatic processes of the host cell. Certain proteins, expressed during both latent and active phases, have the ability to modulate immune surveillance and the molecular pathways associated with tumor suppression. Emerging evidence suggests that human hCMV, whether in a latent or active state, encodes proteins that can influence cellular mechanisms and may contribute to the process of oncogenesis [58,59].

#### 2.5.3. hCMV VIL10 and Malignance

The primary targets of vIL-10 are cell surface proteins, which disrupt effective antigen presentation, costimulation, and adhesion. Furthermore, vIL-10 induces apoptosis by inhibiting antiapoptotic pathways and decreasing the activity of matrix metalloproteinases, leading to interference with cellular communication and biological processes, ultimately enhancing malignancy [6,45,46,60]. During viral infections and protein synthesis in the cytoplasm, the infected cells are targeted for degradation and presented by MHC class I molecules. This mechanism could explain why hCMV vIL-10 targets IL-10 production and downregulate MHC class I and II molecules in immune cells such as monocytes, macrophages, and dendritic cells [61,62,63,64,65].

Viral surveillance and the spread of cancerous cells are interplay mechanisms between tumors and viruses. This leads to an optimal microenvironment for viruses to replicate and at the same time increases the tendency of malignant cells not to be targeted and destroyed by immune cells [66,67,68]. This is suggested as a major factor where viruses contribute to cancer development. Around 15–20% of human cancers are caused by viral infections. Viruses such as Epstein–Barr (EBV), hepatitis virus B (HBV), human papillomavirus (HPV), hepatitis virus C (HCV), Kaposi’s sarcoma-associated herpesvirus (KSHV or HHV8), human T-lymphotropic virus-1 (HTLV-1), and Merkel cell polyomavirus are considered oncogenic because many proteins produced during their replication cycles have been related to different types of cancer [40,67,69,70,71,72]. Recent studies have indicated that viral genetic material can encode a subset of proteins that contribute to “oncogenic hits” [71,73]. These hits represent an accumulation of somatic alterations caused by mutations. This process results in the gradual acquisition of characteristics that lead to a malignant phenotype, often via exposure to carcinogenic environments, immunosuppression, or chronic inflammation [72,74,75]. In many cases, the proteins produced during the latent phases of viral infection are considered potential triggers for cancer development. Although the specific mechanisms remain unclear, it has been reported that certain pathways can be activated following an infection, which may lead to cell transformation [68,71,76]. Currently, hCMV has not been classified as an oncovirus because there is insufficient information regarding the role of its latent-phase proteins in cancer development. However, it has been suggested that the upregulation of interleukin-10 (IL-10) may be how hCMV evades recognition by immune cells and sustains a subdued immune response. During an acute hCMV infection, an inflammatory profile characterized by the production of cytokines and chemokines emerges as a result of early immune responses. This suggests that hCMV, through the action of viral interleukin 10 (vIL10), may indirectly act as a trigger for oncogenesis. Nevertheless, additional studies are necessary to verify the role of this mechanism in the development of disease [59,77,78,79]. Viral cancers resulting from persistent infections can emerge many years or even decades after an initial acute illness. Whether a virus is classified as oncogenic depends on its alignment with a cancer hallmark model, which is a valuable framework for understanding human virus-associated carcinogenesis. Viruses have coevolved with their hosts and have developed crucial abilities to manipulate cellular processes. These include promoting antiapoptotic responses, uncontrolled cell proliferation, and mimicking proteins to activate and modulate specific cellular responses. Additionally, they can cause DNA damage and genetic instability, which lead to increased mutation rates, chromosomal alterations, and chronic inflammation [68,69,72,76]. These hallmarks are relevant factors in the carcinogenesis process, and only a few viruses fall into this category. This is why certain researchers propose that hCMV should be included in this group [80,81]. Cell transformation induced by hCMV may differ from that caused by other DNA viruses. Some studies indicate that viral DNA sequences are not retained in host cells, suggesting that they do not integrate into the host’s genetic material to directly induce oncogenesis. Instead, an alternative mechanism has been proposed involving cell activation and the deregulation of the cell cycle, which may lead to lesions in cellular DNA and potential genomic damage. The functions of hCMV viral proteins are still being elucidated, but they appear to be linked to chromosomal alterations and the transformation of various cellular genes. This includes interactions with proto-oncogenes and tumor suppressor genes, resulting in altered activity of host-encoded proteins and the emergence of an oncogenic phenotype [47,82]. The mechanisms through which human hCMV contributes to oncogenesis are not yet fully understood. However, various studies have suggested that DNA damage occurs in specific regions that regulate growth inhibition and apoptosis, resulting in uncontrolled cell growth. This leads to genetic instabilities that induce mutations linked to viral immediate-early (IE) genes and latent genes [79,80]. Infected cells, whether through active or latent infection, fail to repair themselves effectively because of the viral control mechanisms at work. As a result, DNA lesions can be transmitted during cell division, giving rise to an oncogenic cell line. Many of the mutations associated with hCMV infection affect tumor suppressor genes such as p53 and Rb, which play vital roles in cellular damage responses and in activating cell cycle arrest to facilitate DNA repair and apoptosis. p53 is central to these pathways, as it binds to specific DNA sequences and transactivates target genes. In noninfected cells, p53 is activated by potentially harmful events, including UV or gamma irradiation, extreme heat, hypoxia, starvation, or viral infections [83,84,85]. Research has shown that hCMV induces elevated levels of p53 and hyperphosphorylated Rb. It has been proposed that the viral genome contains 21 exact matches for the p53 DNA binding site, which may include regions that could influence viral protein expression [85,86]. One proposed mechanism by which hCMV interacts with the p53 protein involves its sequestration in the cytoplasm. This prevents p53 from performing its role in promoting apoptosis, as it must be located in the cell nucleus to be active [87,88,89]. Furthermore, hCMV may induce the overexpression of cyclins and result in the formation of hyperphosphorylated retinoblastoma (Rb) protein, which can disrupt cyclin function and interfere with cell cycle regulation. There are two primary mechanisms through which hCMV may promote oncogenesis. The first involves the induction of cellular abnormalities due to active viral replication and protein synthesis during both the lytic and latent phases [83,85,86]. The second mechanism is known as “hit and run”, where hCMV initiates cellular transformation, allowing the virus to be eliminated while the transformed state persists. This state is maintained by the accumulation of mutations that enable the tumor to grow and thrive independently of the virus (Figure 3) [90,91,92,93,94].

### 2.6. IL-10 and vIL10 in hCMV Infection

IL-10 is an anti-inflammatory cytokine produced by a variety of myeloid and lymphoid cells, including monocytes, T cells, macrophages, dendritic cells, natural killer (NK) cells, and epithelial cells. An infection caused by a single pathogen can stimulate the secretion of IL-10 from multiple cell populations, depending on factors such as the type of pathogen, the infected tissues, and the timing within the immune response [92,93]. IL-10 exhibits immunosuppressive properties that safeguard the body against excessive inflammatory responses, which can lead to organ damage. Additionally, it plays a role in regulating immune cell activation and reducing the production of proinflammatory cytokines. Some viruses exploit this mechanism to evade antiviral responses and establish lifelong infections. hCMV not only regulates IL-10 production but enhances its expression during both the lytic and latent phases, rendering immune responses less effective and potentially facilitating the reactivation and proliferation of malignant cells.

Numerous viruses encode counterparts of human IL-10 known as viral IL-10s (vIL-10s) or virokines. These viral proteins are thought to have been acquired from their host during evolutionary adaptations [45,61,62]. Advances in sequencing technologies have identified an expanding array of vIL-10s across various viral families, including twelve from the *Orthoherpesviridae* family, two from the *Alloherpesviridae* family, and seven from the *Poxviridae* family [45]. It is proposed that recombination between a viral genome and the host genome during viral replication can lead to the acquisition of host genes, as well as recombination involving the viral genome and retrotranscribed cDNA derived from mRNA. The adaptive evolution of vIL-10 via positive selection has preserved the characteristics most advantageous for the viral life cycle, enhancing the strategies for latency and manipulation of host cell machinery [20]. Selective pressure may induce the shortening or loss of one or more introns in specific genes, which illustrates the diverse vIL-10 isoforms identified in various studies as well as their distinct functions [45,67,94]. Despite differences in sequence homology between IL-10 and vIL-10 proteins, their structures are remarkably conserved, enabling the virus to effectively mimic these properties. Research indicates that the genes encoding vIL-10s are transcribed at varying times during in vitro replication, with hCMV vIL-10 being expressed at later stages [43,45,95]. Although vIL-10 genes are not essential for the growth of hCMV, they are crucial for receptor binding. While only a limited number of studies have examined the role of vIL-10 in vivo, it is proposed that it may contribute to the recruitment of permissive cells at sites of viral replication, thereby facilitating the spread of infection and latency within the host. The hCMV viral IL-10 protein consists of 175 amino acids and, according to the reports, exhibits only 27% homology with human IL-10. We conducted an analysis of IL10 and vIL10 from the reference strains Towne, Toledo, AD169, and Merlin. The access codes for these strains are as follows: CR541993.1, FJ616285.1, GU937742.2, FJ527563.1, and NC_006273.2. Additionally, we examined sequences of various reported vIL10 isoforms, with access codes OP346856.1, OP346856.1 OP346857.1, OP346858.1, OP346859.1, and OP346860.1 OP346861.1. An alignment of the DNA sequence was performed using the Clustal W2 program (available online at: https://www.ebi.ac.uk/jdispatcher/, accessed on 8 December 2024) (Appendix A).

Identity percentages were calculated using the JalView program (also available online). Our findings revealed that the identity between human IL10 and hCMV vIL10 displayed a similarity of 40% to 50% across the different reference strains and isoforms (Table 2). These findings indicate that specific hCMV strains may confer differential risks for leukemia development, necessitating further investigation to enrich previous reporting [96].

In contrast, the identity of vIL10 evaluated among various strains and isoforms ranged from 67% to 89% (Table 3). This indicates that although these proteins share the same function and origin, the relatively low identity may be linked to some form of selection mechanism that does not compromise their functionality. To fully assess this, it would be necessary to measure the level of functionality in wild strains.

Despite this limited similarity, vIL-10 effectively binds to its receptors, ILR1 and ILR2, and disrupts several cellular processes, including the cell cycle, cell proliferation, differentiation, and apoptosis. This protein has been thoroughly characterized at both functional and structural levels during the lytic phase, playing a significant role in IL-10 regulation in latent cells. Throughout the infectious process, alternative splicing produces different isoforms of this gene, with two primary protein isoforms identified: vIL-10 (or cmvIL-10) and LavIL-10 [43,45,67,97,98]. At least five vIL-10 isoforms resulting from alternative splicing have been documented during hCMV replication, although their specific functions remain largely unexplored. Nonetheless, it is believed that both vIL-10 and LavIL-10 are involved in the downregulation of MHC I and II molecules, which may disrupt the STAT3 signaling pathway—a critical factor in various cellular processes and a potential contributor to cellular transformation [99,100]. The regions on the surfaces of IL-10 and its vIL-10 variants that engage with receptors are fundamentally similar. This resemblance enables human cytomegalovirus (hCMV) to signal through the host’s IL-10 receptor (IL-10R) in a manner comparable to that of IL-10 itself. Consequently, the virus can activate essential signaling pathways involving Janus kinase 1 (Jak1), tyrosine kinase 2 (Tyk2), and STAT transcription factors, which are critical for the transcription of host genes. The most significant structural difference between IL-10 and vIL-10 lies in the interdomain angle. In order to facilitate binding, a reorganization of the IL-10R1 subunit within the cell surface complex is necessary [101]. Recent studies have indicated that viral IL-10 (vIL-10) plays a significant role in influencing monocyte polarization, particularly by promoting the development of M2c alternatively activated monocytes. This process facilitates the upregulation of the anti-inflammatory enzyme heme oxygenase-1 (HO-1), which is crucial for vIL-10-mediated suppression of proinflammatory cytokine profiles by M2c monocytes, as well as for reducing TCD4+ cell stimulation [102]. vIL-10 also inhibits antigen presentation by downregulating MHC I molecules while simultaneously protecting certain cells from NK cell lysis through the upregulation of human leukocyte antigen G (HLA-G) [16,36]. Additionally, vIL-10 hinders the maturation of dendritic cells (DCs), resulting in a reduced immune response that adversely impacts their functionality and survival (Figure 4) [103,104].

An analysis of the geographic distributions of the hCMV genome has revealed that, despite exhibiting greater genomic variability than other herpesviruses, hCMV remains highly conserved. This conservation is associated with a selective equilibrium influencing the immunomodulatory regions. From a genetic perspective, this suggests that the maintenance of allele frequencies remains consistent, independent of geographic location [105]. We observed substantial variability among the various isoforms of vIL10. Nevertheless, there is no evidence to suggest that the functionality of the protein is compromised in this context. It is imperative to consider additional factors pertinent to the studied population, such as ethnicity, susceptibility, and associated mutations, as these may potentiate the tumorigenic potential of vIL10 in oncogenesis.

## 3. The Role of the Immune System in Tumorigenesis

An effective immune response is fundamental for the regulation of tumorigenesis. In acute lymphoblastic leukemia (ALL), immune cells play a pivotal role in both innate and adaptive immunity. Within the innate immune response, natural killer (NK) cells are critical for the identification and elimination of malignant cells that lack MHC class I molecules. NK cells contribute not only to antiviral defense but to tumor surveillance. Notably, NK cell responses to viral infections can exert a significant influence on tumor dynamics. Viral infection leads to the hijacking of the host cell’s machinery, thereby altering cellular function. The interplay between viral activity and microenvironmental factors ultimately dictates the fate of infected cells. Furthermore, macrophages and dendritic cells facilitate tumor elimination through phagocytosis, subsequently processing and presenting tumor-associated antigens via MHC class I molecules. In the adaptive immune response, CD8+ T cells recognize these antigens and mediate the apoptosis of leukemic cells. Additionally, CD8+ T cells contribute to the production of tumor-targeting antibodies, which enhance the opsonization and subsequent elimination of malignant cells by macrophages and NK cells [106,107,108].

## 4. The Failure of NK Cells and Tumorigenesis

When the immune system successfully manages the viral infection, clearance of the infected cells is expected. However, deficiencies in the immune response can occasionally lead to the development of malignancies. Various mutations can affect the quantity and functionality of NK cells. Some of these mutations impair the lytic function of NK cells, resulting in cytotoxic dysfunction. Such mutations can hinder NK cells’ ability to eliminate virally infected cells and diminish their antitumor activity. However, the immune response mediated by NK cells can be affected during the viral infection as follows.

(a) Molecules Involved in Signaling Pathways. Mutations on the phosphoinositide 3-kinase (PI3K) pathway are vital for NK cell cytotoxic activity. The inhibition of PI3K110d, a subunit of PI3K, leads to a reduction in NK cell numbers due to impaired maturation, which also results in decreased cytotoxic function. These observations confirm that disruption of signaling through PI3K110d can compromise NK cell function. Previously identified mutations of this isoform have been associated with viremia from hCMV and Epstein–Barr virus (EBV), as well as lymphoid hyperplasia (Figure 5) [109].

The coiled-coil domain-containing 22 (CCD22) gene encodes a protein that plays a role in activating nuclear factor (NF)-kB via signaling through the interleukin-2 receptor (IL-2R) [110]. This activation regulates perforin expression in natural killer (NK) cells (Figure 6) [111].

(b) Mutations in receptors. An aggressive leukemia of NK cells (ALNKC) has been associated with Epstein–Barr virus (EBV) infection. A previous case report detailed a patient with ALNKC who initially presented with hemophagocytic lymphohistiocytosis and had a mutation in the CCD22 gene. This mutation may have affected perforin expression in NK cells and could be implicated in both conditions [112]. This case suggests the need for further studies to explore whether EBV is involved in the development mechanism of ALNKC or if its presence is merely an artifact linked to viral reactivation during a critical stage of the patient’s immune response, which complicated the control of the viral infection. Natural killer deficiencies (NKDs) are primary immunodeficiencies that impact the CD56+CD3+ cell lineages, which are responsible for mediating antitumor cytotoxicity. NKDs can impair NK cell activity and facilitate the reactivation of herpesviruses such as EBV, which can lead to tumorigenesis [113].

(c) Mutations in transcription factors. Mutations in transcription factors, including GATA2, RTEL1, GINS1, IRF8, and MCM4, have been identified as causes of classical NKD (CNKD). Additionally, mutations in the CD16 gene are associated with functional NKD (FNKD) [114,115]. STAT3, a transcription factor that is part of the STAT family and involved in the interferon response, has been connected to antiviral responses, including those against influenza and vaccinia. For instance, knocking down STAT3 enhanced replication of influenza and vaccinia viruses, highlighting the crucial role of STAT3 in controlling these viral infections [116,117]. Mutations in STAT3 also influence tumor proliferation; inhibitors of STAT3 can alter the susceptibility of leukemia cells to apoptosis, while normal cells remain unaffected. This indicates a shared mechanism by which mutations impair NK cells’ ability to manage viral replication and control tumor growth [118,119,120].

(d) Chronic active infections of EBV (CAEBV). CAEBV (chronic active Epstein–Barr Virus) can diminish NK cell antiviral function as well [119,120]. This type of infection is recognized as a lymphoproliferative disorder that presents with a broad spectrum of clinical manifestations. CAEBV infection are particularly predisposed to the development of leukemia [118]. It has been shown that both T and NK cells in CAEBV patients exhibit intragenic deletions common in neoplastic disorders, leading to a predisposition for NK or T-cell leukemia or lymphoma [117]. In CAEBV, mutations are linked to the clonal evolution of multiple cellular lineages of lymphoid progenitors infected with EBV, some of which are associated with NK/T-cell lymphomas. Notably, mutations are not found in less severe EBV-related conditions such as infectious mononucleosis or lymphoproliferative syndrome, suggesting a significant role for both EBV infection and mutations in the development of lymphoma and potentially leukemia [121]. Viral infections can also stimulate NK cell proliferation. This phenomenon was first observed in T cells infected with human T-cell leukemia virus type 1 (HTLV-1), which has a well-documented association with leukemia. HTLV-1 infection is linked to spontaneous high levels of NK cell proliferation and expansion of CD56-expressing NK cells [122]. Furthermore, it has been noted that HTLV-1 can activate KIR3DL2, an inhibitory receptor prevalent in patients with leukemia, which facilitates the evasion of leukemic cells from NK cell immunity. Recent studies have suggested a potential association between active hCMV infection and an increased risk of developing acute lymphoblastic leukemia (ALL) [123]. While chronic human hCMV infection is prevalent, it does not consistently correlate with the manifestation of severe disease in congenital cases. Nevertheless, congenital hCMV infection has been identified as a potential risk factor for the development of leukemia [124,125]. In older women, chronic hCMV infection has also been associated with frailty, a clinical syndrome frequently observed in the aging population, characterized by systemic inflammation. Furthermore, chronic hCMV infection has been implicated in chronic rejection among transplant recipients [126]. It is noteworthy that, in these contexts, inflammation—often associated with cytokine imbalances—may play a crucial role. Within this framework, it becomes essential to investigate the contribution of inflammation, particularly that induced by proteins derived from either active or latent hCMV infections, to the pathogenesis of ALL. However, the strength of this association is contingent upon the individual patient’s physiological and immunological profile.

## 5. The Influence of Altered Macrophages in the Leukemic Microenviroment

Leukemic cells have developed multiple sophisticated mechanisms to circumvent the host’s immune surveillance. Among these mechanisms are:

(a) Alteration of Phagocytic Signals. Within the tumor microenvironment, macrophages and dendritic cells lose their ability to effectively recognize leukemic cells. Several mechanisms have been described, including a phenomenon in which specific subtypes of M2 macrophages display elevated levels of markers associated with prosurvival signals. These markers disrupt the function of the “do not eat me” signal, a critical mediator of phagocytosis, thereby inhibiting the immune cells’ capacity to recognize and engulf leukemic cells [127].

(b) Induction of an Immunosuppressive Microenvironment. Within the tumor microenvironment, macrophages typically play a crucial role in orchestrating immune responses and facilitating the clearance of tumor cells. However, when these macrophages undergo polarization to the M2 phenotype, they actively contribute to the establishment of an immunosuppressive microenvironment. This polarization not only impairs the immune response but fosters an environment that promotes the proliferation, survival, and metastasis of leukemic cells [128].

## 6. Influence of the Altered Acquired Immune System in the Leukemic Microenvironment

Several mechanisms contribute to the dysfunction of CD4+ and CD8+ T cells within the leukemic microenvironment, including:

(a) Immunosenescence. This phenomenon refers to the premature aging of T cells, which leads to a diminished capacity for the elimination of malignant cells. As T cells age, they accumulate cellular alterations that limit their ability to proliferate and respond to antigenic stimuli effectively. This decline in function is further exacerbated by changes in cytokine production, which impair immune response efficacy [129].

(b) Immune Evasion. This mechanism is of particular significance in the context of leukemia, as leukemic cells possess the ability to evade detection by the host’s immune system. Leukemic cells can downregulate or completely lose the expression of antigens recognized by T cells, thus avoiding immune surveillance. Additionally, these cells express the PD-L1 (programmed death ligand-1) receptor, which binds to the PD-1 receptor on T cells, effectively inhibiting their activation. Moreover, leukemic cells secrete various cytokines that suppress T-cell activity, further contributing to immune evasion [127].

(c) T-Cell Inflation. This mechanism is characterized by a prolonged and dysregulated immune response, wherein T cells are persistently activated by inflammatory cytokines released by leukemic cells. These cytokines not only promote resistance to apoptosis but enhance cellular proliferation, disrupting the delicate balance of cytokines necessary for optimal T-cell function. As a result, this altered immune landscape contributes to the failure of effective immune surveillance and the persistence of the malignant cells [130].

The mechanisms described is mediated by hCMV in infected cells, thereby indicating a significant association between hCMV and leukemia.

## 7. The Antileukemic Potential of hCMV

From another perspective, hCMV could potentially act as an antileukemic agent. This phenomenon has been observed in patients experiencing reactivation of hCMV, which is common among those who have undergone bone marrow transplants (BMTP). While the general consensus views such reactivation as harmful, evidence suggests that it may provide a protective effect for BMTP patients. In vitro studies have shown a proapoptotic effect on leukemic cells infected with hCMV [131,132,133]. Recent analyses propose that natural killer (NK) cells play a crucial role in this protective effect. During hCMV infection, there is an expansion of NK cells that leads to increased expression of the HLA-E receptor NKG2C. This expanded NK cell population is characterized by NKG2C+/NKG2A− cells, which may enhance their ability to combat leukemic cells [134]. Additionally, an increased number of CD56bright CD16dim/DNAM1+ NK cells have been observed, indicating that these cells may contribute to the antileukemic effects associated with hCMV infection [132]. Moreover, oncolytic activity induced by herpesvirus type 1 in other tumors has been reported in cases of leukemia, with this cytolytic activity being facilitated by NK cells. This process involves a synergy between viral surface components and interleukin-2 and interleukin-15, which activate the cytolytic capabilities of NK cells [135]. A previous study examined the effects of intravenous coxsackievirus A21 (CAV21) on peripheral blood mononuclear cells from healthy individuals and leukemia patients. The study found that CAV21 activates NK cells through immunological mechanisms, which include the activation of the type I interferon response mediated by ICAM-1 and plasmacytoid dendritic cells. These findings suggest that CAV21 may serve as a promising immunotherapeutic agent [136]. Furthermore, hCMV infection induces the expression of high levels of CD94/NKG2C and a specific ligand for CD94/NKG2C on the surface of NK cells [137], indicating that hCMV infection can modulate the repertoire of NK cell receptors [138]. However, hCMV reactivation has also been reported in patients with Philadelphia-positive leukemia undergoing treatment with dasatinib, a tyrosine kinase inhibitor [139]. Notably, the benefits of this reactivation were evident; lymphocytosis was associated with an increased number of large granular lymphocytes enriched with NK cells following dasatinib treatment. This suggests that hCMV reactivation is essential for NK cell activation [140]. Additionally, increased NK cell maturation was observed in a patient who received umbilical cord blood transplants, characterized by the expansion of CD56dim NKG2A− killer cells and the deletion of NKG2C [141]. Further research is necessary to understand the mechanisms through which hCMV reactivation could serve as a therapeutic approach against leukemic cells.

## 8. Conclusions

The interaction between host cells and human hCMV is intricate and not yet fully understood. Continued research is essential to clarify the mechanisms that regulate hCMV latency and reactivation, as well as the proteins involved in these processes as well as different molecules that can be part of the cellular functions. This understanding will illuminate both the potential role of hCMV in leukemogenesis and its antileukemic properties. A growing body of evidence suggests that vIL-10 may indirectly influence leukemogenesis by disrupting immunological pathways. The potential for different strains of hCMV to exhibit varying levels of risk cannot be overlooked, thus highlighting the necessity for further research. It is imperative to interpret the reported leukemic and antileukemic effects with caution, as these may be contingent upon the timing of host–virus interactions and the intricate biological cycles of both the host and the virus. A comprehensive understanding of these interactions is crucial for elucidating the potential role of hCMV in leukemia development and for identifying novel therapeutic targets.

## Figures and Tables

**Figure 1 viruses-17-00435-f001:**
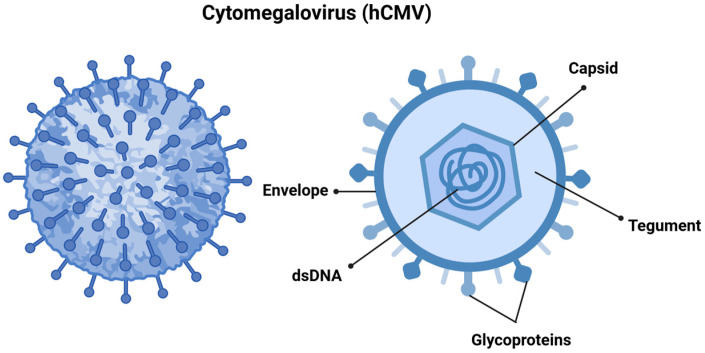
Cytomegalovirus is structured with double-stranded DNA (dsDNA) encased within an icosahedral nucleocapsid. This nucleocapsid is surrounded by a tegument layer made up of viral proteins, all of which are enveloped in a membrane derived from the host cell. This envelope contains structural glycoprotein complexes that play a crucial role in the attachment and recognition of host cells. Created in https://BioRender.com. Accessed on 8 December 2024.

**Figure 2 viruses-17-00435-f002:**
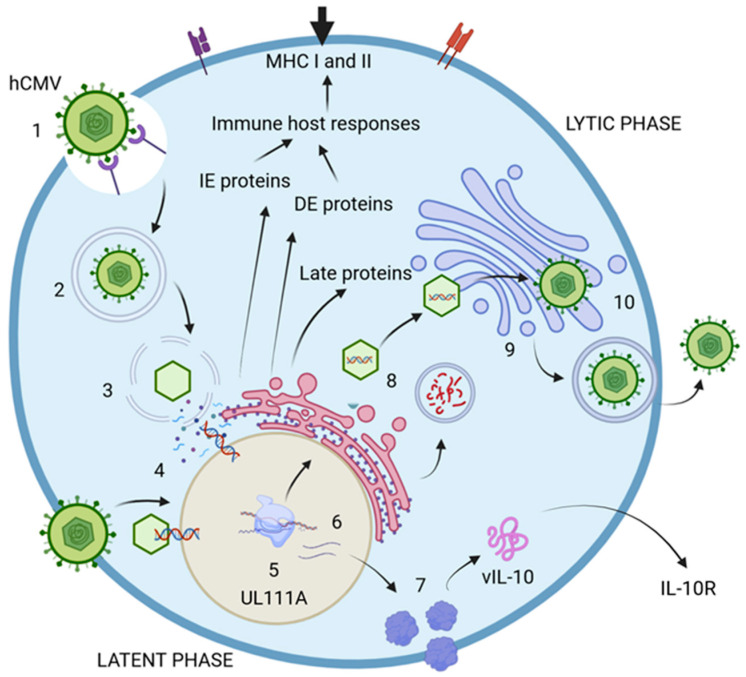
hCMV life cycle. 1. The virus initially attaches to cell receptors and enters the host cell, either through the endocytic pathway or via direct fusion with the cell membrane. 2. The endosome then transports the virion to the cytosol. 3. The viral nucleocapsid is translocated into the nucleus, where the viral double-stranded DNA is released. 4. The expression of immediate-early proteins (IE-1 and IE-2) commences, followed by the production of delayed-early proteins (DE). The DNA undergoes replication to generate new viral particles, while late proteins (L) are synthesized to facilitate the replication process. 5. During the latent phase of infection, viral particles are not actively produced; however, certain proteins continue to be expressed. An example is the UL111A gene, which encodes vIL-10, a virokine and homolog of IL-10. 6. Viral mRNA is translated into various proteins, including vIL-10. 7. vIL-10 interacts with and modulates the IL-10 receptor (IL-10R), thereby influencing cytokine production. 8. The DNA, once assembled in capsids, is transported to the endoplasmic reticulum (ER) for envelopment and is subsequently moved to the Golgi apparatus for the addition of glycoproteins and envelopes. Numerous proteins are encoded to assemble new virions. 9. The fully assembled virions are finalized in the Golgi apparatus and then delivered to the cell surface. 10. Mature virions exit the infected cell through the process of exocytosis. Created in https://BioRender.com. Accessed on 8 December 2024.

**Figure 3 viruses-17-00435-f003:**
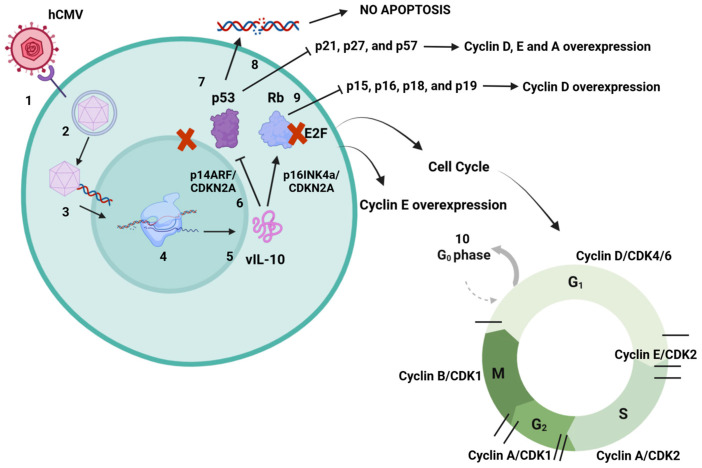
hCMV dysregulation mechanisms of p53 and Rb. 1. hCMV binds to cell surface receptors and enters the cell. 2. Capsid is delivered to the nucleus. 3. Capsid releases viral DNA into the nucleus, producing proteins to assemble new virions. 4. During the latent phase of infection, many proteins are encoded to help immune evasion. 5. mRNAs are translated into a set of proteins that includes vIL-10. 6. It has been suggested that vIL-10 may inhibit p14ARF and p16INK4a, both proteins that control tumor suppression through p53 and Rb, leading also to p53 and Rb inhibition. The mechanism is still unknown, but vIL-10 may sequester p53 in the cell cytoplasm and inhibit its function because of its specific activity in the cell nucleus, causing many cell dysregulations. 7. Inhibition of p53 prevents damaged DNA from inducing cell apoptosis, which may lead to error duplication. 8. Nonfunctional p53 inhibits the activity of p21, p27, and p57, proteins with tumor suppressor functions. This may lead to cyclin D, E, and A overexpression related to oncogenesis. 9. Nonfunctional Rb inhibits the activity of p15, p16, p18, and p19. These proteins control cyclin D, with their inhibition leading to overexpression of cyclin D, also related to tumorigenesis. 10. Interaction between Rb and E2F cell factor inactivates E2F. Normally, cyclins D and E phosphorylate Rb, causing its dissociation. Nevertheless, Rb inhibition causes more E2F activity and cyclin E overexpression, which may lead to cancer development. Both of the proteins p53 and Rb have major roles in cell cycle progression. Created in https://BioRender.com. Accessed on 26 February 2025.

**Figure 4 viruses-17-00435-f004:**
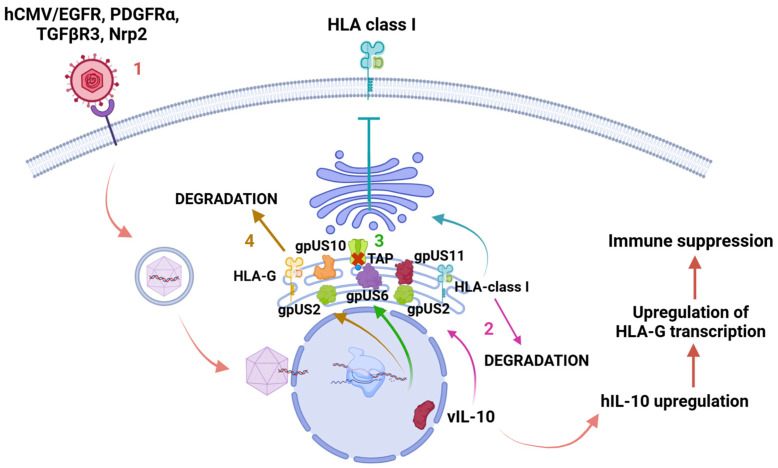
Human leukocyte antigen-G (HLA-G) downregulation. 1. When hCMV enters the cell through receptors or direct membrane fusion, the capsid is delivered to the nucleus, where many proteins are encoded during the lytic and latent phases. Protein vIL-10 is encoded during the latent phase, causing an upregulation of hIL-10 and human leukocyte antigen G (HLA-G) transcription, resulting in immune suppression. 2. Proteins gpUS2 and gpUS11 redirect HLA class I from the endoplasmic reticulum (ER) to cell cytosol and induce its degradation. 3. GpUS6 affects antigenic presentation by inhibiting the associated transporter function (TAP), not allowing peptide delivery into the endoplasmic reticulum and MHC class I binding. 4. Proteins gpUS2 and gpUS10 induce HLA-G degradation in cell cytosol after its retention in ER. Created in https://BioRender.com. Accessed 26 February 2025.

**Figure 5 viruses-17-00435-f005:**
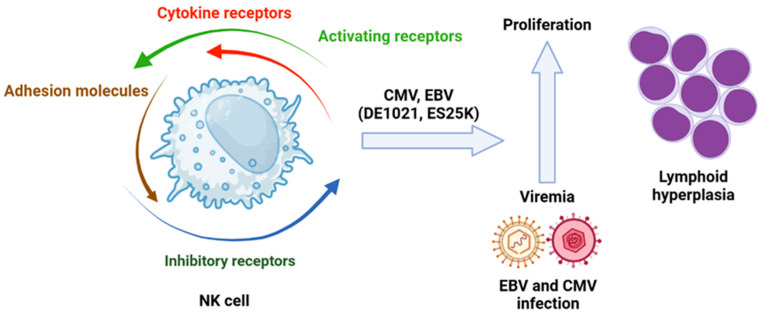
Role of viral infection in leukemogenesis in which the NK cells are involved. Isoforms of PI3K110δ (DE1021K and E525K) associated with hCMV or EBV infection have an impact in lymphoid hyperplasia. Created in https://BioRender.com. Accessed on 26 February 2025.

**Figure 6 viruses-17-00435-f006:**
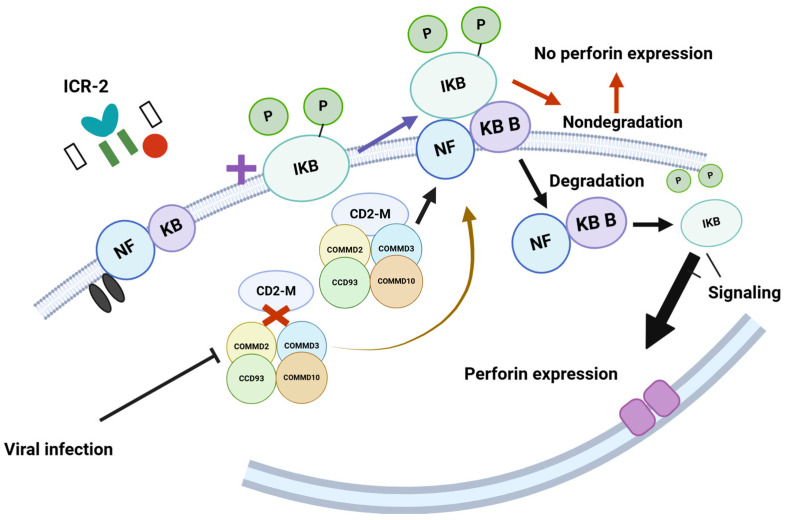
The model proposes that COMMD2, COMMD3, CCD93, and COMMD10 interact with CD22M to form a multimer. This multimer then interacts with phosphorylated NFKB/iKB, leading to its degradation, which facilitates the signaling necessary for perforin expression. However, viral infections disrupt the interaction between CD22M and the proteins COMMD2, COMMD3, CCD93, and COMMD10. This disruption prevents the degradation of NFKB/iKB and negatively impacts perforin expression. Such mechanisms are crucial for recognizing virus-infected cells; when this recognition process is compromised, it hinders the immune response. https://BioRender.com Accessed 26 February 2025.

**Table 1 viruses-17-00435-t001:** Some known mechanisms of immune evasion associated with hCMV.

**PROTEIN**	**IMMUNE EVASION MECHANISMS**	**REFERENCES**
gB	Interaction with cell integrins. Tropism and the Facilitation of Complete Infection. Entry into fibroblasts and epithelial cells. Formation of a pentameric complex with gH and gL; improves viral entry in epithelial, endothelial, monocytic and dendritic cells, which relates to infection levels. The processes of viral cell attachment and replication. Retention of MHC class I and reduction in MHC class II expression. Promotion of the degradation of MHC class I. Promotion of MHC I degradation. Retention of MHC class I heavy chains and the induction of HLA-G degradation. Inhibition of TAP and the process of peptide translocation. MHC Class I proteins experience a delay when transporting from the endoplasmic reticulum (ER) to the Golgi apparatus. Prevention of the formation of viral antigenic peptides and the response to interferons.	[21]
gH/gL		[21]
gO		[21]
UL128, UL130, and UL131a		[21]
gM/gN		[21]
US3		[22,23,24,25]
US11		[26,27]
US2		[28,29]
US10		[30,31]
US6		[32,33]
pp71		[34]
pp65		[35]

**Table 2 viruses-17-00435-t002:** Percentage of identity of IL10 compared with vIL10 from reference strains and their isoforms.

	TOWNE	TOLEDO	AD169	MERLIN	56.1	57.1	58.1	59.1	60.1	61.1
hIL-10	42.75	41.28	40.70	41.44	46.0	51.17	49.60	48.54	50.30	49.14

**Table 3 viruses-17-00435-t003:** Percentage of identity of vIL10 between reference strains and their isoforms.

	TOWNE	TOLEDO	AD169	MERLIN	56.1	57.1	58.1	10 59.1	10 60.1	61.1
TOWNE	100	89.39	83.09	83.48	87.07	68.47	78.44	59.01	62.21	68.98
TOLEDO		100	83.03	82.87	86.31	67.31	77.30	57.98	61.05	67.95
AD169			100	82.99	86.44	67.57	77.30	58.24	61.18	67.95
MERLIN				100	87.07	68.48	78.06	58.72	61.70	70.08
56.1					100	73.17	83.62	63.36	66.58	73.44
57.1						100	84.33	80.95	89.15	77.30
58.1							100	70.34	76.84	82.12
59.1								100	77.18	82.59
60.1									100	72.87
61.1										100

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
