# Peer review of "Is the vIL-10 Protein from Cytomegalovirus Associated with the Potential Development of Acute Lymphoblastic Leukemia?"

_viruses, 2025, doi:10.3390/v17030435_

Round 1

Reviewer 1 Report

Comments and Suggestions for Authors

vIL-10, CMV and leukemia

this is a review of mechanisms by which HCMV may induce or impact the genesis of leukemia and other cancers.  It is a nice description of the variety of mechanisms and physiologic disruptions that hCMV can produce to influence tumorigenesis, possibly; concentrating on the functions of vIL-10. There is no attempt to review the epidemiology of hCMV and leukemia, but that is not the focus here and so is appropriate.

First of all, CMV’s association with leukemia is only with pediatric acute lymphoblastic and not other types of leukemia – childhood ALL is very rare compared to adult leukemias, so this should be specified in the abstract and introduction. That is as far as I will critique the epidemiology!

Section 2 – this section reviews some mechanisms about viral infections having an impact on NK function and presumably tumor immunosurveillance by NK cells. The authors might change the title of this section as the current one only fits obliquely. Also, I would suggest placing this section later in the manuscript as it distracts the reader from the main focus (of the manuscript) on hCMV, and vIL-10 which are buried beneath this section.  Perhaps it could be placed before the conclusion section, or removed entirely. The conclusion does not incorporate or refer back to this section at all, so it could be removed (or a sentence added to the conclusion that incorporates this section).

The extended paragraph that covers pages 8 and 9 is key, and well explains the nuances of “oncogenic” viruses and where hCMV sits in comparison with others. This paragraph might be split into 3 to cover different topics rather than one extended paragraph.  For instance, one to explain true oncogenic viruses, another to contrast and compare hCMV to these others, and a third to point at hCMV factors that influence other oncogene or tumor suppressor function (RB, TP53, etc)

The term “oncomodulatory” is often used rather than “oncogenic” for hCMV and might be defined and explained in this part of the paper.  Or incorporated into the vIL-10 section.

The various vIL-10 isoforms is an interesting section but it is somewhat unsatisfying that this diversity of isoforms and sequence variants exists without conclusive nuance. Is vIL-10 variation related to distribution of the virus geographically, or with particular virulent strains, or putatively oncogenic variants of hCMV? These questions are likely unanswerable with current data, and the authors might add a paragraph or section on “what are the gaps in understanding that must direct the focus of hCMV and vIL-10 related research”

Minor points – some of the figures seem to use “Biorender” program, should this be cited?

Figure 6 is helpful is probably too much detail, and could be placed in a supplement and/or displayed pictorially only (replacing mismatches and gaps with colors and lines, resulting in a one line smaller figure)

Author Response

Dear Reviewer. The authors of the paper “Is the vIL10 Protein Implicated in Potential Leukemogenesis Induced by Cytomegalovirus?” sincerely appreciate your insightful observations and the time you devoted to reviewing our work. In this revised version, we have organized the content to better enrich our contribution to the field and have included additional references to strengthen our objectives. We have thoughtfully addressed each of your comments as follows:

  1. CMV’s association with leukemia is only with pediatric acute lymphoblastic and not other types of leukemia – childhood ALL is very rare compared to adult leukemias, so this should be specified in the abstract and introduction. That is as far as I will critique the epidemiology!

Thank you for your observation. We have included a section on epidemiology.

  1. Section 2 – this section reviews some mechanisms about viral infections having an impact on NK function and presumably tumor immunosurveillance by NK cells. The authors might change the title of this section as the current one only fits obliquely. Also, I would suggest placing this section later in the manuscript as it distracts the reader from the main focus (of the manuscript) on hCMV, and vIL-10 which are buried beneath this section. Perhaps it could be placed before the conclusion section, or removed entirely. The conclusion does not incorporate or refer back to this section at all, so it could be removed (or a sentence added to the conclusion that incorporates this section).

Thank you for your insightful observation. In response, we have reorganized and repositioned this section as complementary information. We believe it is crucial for understanding the immunological mechanisms that are impacted in leukemic cells, especially regarding how these effects can be triggered by hCMV infection.

  1. The extended paragraph that covers pages 8 and 9 is key, and well explains the nuances of “oncogenic” viruses and where hCMV sits in comparison with others. This paragraph might be split into 3 to cover different topics rather than one extended paragraph. For instance, one to explain true oncogenic viruses, another to contrast and compare hCMV to these others, and a third to point at hCMV factors that influence other oncogene or tumor suppressor function (RB, TP53, etc).

Thank you for your observation. We have reorganized this section to improve comprehension and the quality of information distribution.

  1. The term “oncomodulatory” is often used rather than “oncogenic” for hCMV and might be defined and explained in this part of the paper.  Or incorporated into the vIL-10 section.

Thank you for your recommendation. The term "oncomodulatory" has been defined in the vIL-10 section.

  1. he various vIL-10 isoforms is an interesting section but it is somewhat unsatisfying that this diversity of isoforms and sequence variants exists without conclusive nuance. Is vIL-10 variation related to distribution of the virus geographically, or with particular virulent strains, or putatively oncogenic variants of hCMV? These questions are likely unanswerable with current data, and the authors might add a paragraph or section on “what are the gaps in understanding that must direct the focus of hCMV and vIL-10 related research”

Thank you for your observation. A paragraph regarding geographic distribution has been added to that section.

  1. Minor points – some of the figures seem to use “Biorender” program, should this be cited?
  2. Thank you for your feedback. I have now updated the figures to include a citation for Biorender.

  1. Figure 6 is helpful is probably too much detail, and could be placed in a supplement and/or displayed pictorially only (replacing mismatches and gaps with colors and lines, resulting in a one line smaller figure)

Figure 6 has been redesigned based on the recommendations and is now included as supplemental material.

Reviewer 2 Report

Comments and Suggestions for Authors

In this Review, the authors investigated the role of viral interleukin-10 (vIL-10) produced by human cytomegalovirus (CMV) during its latent phase in potentially contributing to leukemogenesis. Despite not being classified as an oncovirus, CMV's ability to align with cancer hallmark models raises questions about its oncogenic potential. The study focuses on vIL-10's role in cancer-related processes.

The topic of this review is both timely and relevant, as the potential oncogenic properties of CMV have been studied for decades. However, definitive evidence supporting CMV's role in oncogenesis remains elusive. Additionally, some studies suggest that CMV may confer potential oncoprotective properties to its human host, adding a polarized dimension to this ongoing debate.

The manuscript is well-structured, and is easy to comprehend. Some minor text formatting issues exist, but these will be most likely addressed by the MDPI team. I suggest some minor polishing of the English language for more clarity.

I would like to express my gratitude to the Authors for taking the time to consider my comments, which are as follows:

Comments:

·         Lines 61-62: The phrase “…some studies suggest that in utero exposure to hCMV may alter neonatal cytokine levels” in the Introduction must be referenced.

·         Lines 62-32: The phrase “Furthermore, the expression of viral IL-10 by hCMV could modulate the immune response” in the Introduction must be referenced.

·         In the “2. Impact of Immune Response and Tumorigenesis Associated with Viral Infection” section of the Review, CMV is conspicuously mentioned only once; the rest of the text is dedicated to EBV and its relationship with tumorigenesis. I understand that this section serves as a prelude to the main theme of the Review; however, as the work deals with potential CMV-related tumorigenesis, I kindly suggest that the Authors add text related to the link between immunosuppression/immune system and CMV replication within the host.

·         Moreover, lack of T-cell CD8 and CD4 involvement can contribute to viral-induced cancer such as Kaposhi’s sarcoma (which caused Kaposi's Sarcoma-associated Herpesvirus, also a latent virus from the Orthoherpesviridae such as EBV and CMV), so a reference about T-cell role in oncogenesis would be a suggestion from my part.

·         Line 149: The new nomenclature is “Orthoherpesviridae”, not Herpesviridae.

·         It would seem that the acronym “hCMV” is appearing in brackets on multiple occasions within the text. It needs to appear only once, at the beginning (Lines 61,196, 202…).

·         Although it is referenced by three papers, the phrase frol Lines 272-275 is not clear: “Viral surveillance and cancerous cells spreading are interplay mechanisms between tumours and viruses, this leads to an optimal microenvironment for viruses to replicate and at the same time increases malignant cells not to be targeted and destroyed by immune cells”. Can the Authors kindly elaborate what this interplay means? Can the Authors explain in more clear terms what “increases malignant cells” means?

·         Since, as the Authors write, “human IL10 and hCMV vIL10 displays a similarity of 40% to 415 50% across the different reference strains and isoforms”, could it be that only some stains of CMV are potentially cancerogenous, and others are not? I kindly suggest that the Authors add this observation to their work, since high-risk (HR) HCMV strains have been described in literature (Herbein, G. High-Risk Oncogenic Human Cytomegalovirus. Viruses 202214, 2462).

·         In the Conclusion section, the Authors offer a general insight into the thematic; however, I kindly suggest that the paragraph includes a sentence elaborating on the primary evidence supporting (or not supporting) the role of vIL-10. What does the reader keep from the text – is the literature more for or against implicating the vIL-10 in potential leukemogenesis?

After revising as per above, I recommend the Manuscript for publication.

Author Response

Dear Reviewer. The authors of the paper “Is the vIL10 Protein Implicated in Potential Leukemogenesis Induced by Cytomegalovirus?” sincerely appreciate your insightful observations and the time you devoted to reviewing our work. In this revised version, we have organized the content to better enrich our contribution to the field and have included additional references to strengthen our objectives. We have thoughtfully addressed each of your comments as follows:

  1. Lines 61-62: The phrase “…some studies suggest that in utero exposure to hCMV may alter neonatal cytokine levels” in the Introduction must be referenced.

Thank you for your comments; the paragraph has been referenced.

  1. Lines 62-32: The phrase “Furthermore, the expression of viral IL-10 by hCMV could modulate the immune response” in the Introduction must be referenced.

Thank you as well. Similar to the previous paragraph, this phrase was also referenced.

  1. In the “ Impact of Immune Response and Tumorigenesis Associated with Viral Infection” section of the Review, CMV is conspicuously mentioned only once; the rest of the text is dedicated to EBV and its relationship with tumorigenesis. I understand that this section serves as a prelude to the main theme of the Review; however, as the work deals with potential CMV-related tumorigenesis, I kindly suggest that the Authors add text related to the link between immunosuppression/immune system and CMV replication within the host.

Thank you for your observation. This section has been reorganized to present first the information about a virus with proven oncogenic activity, such as EBV, followed by details on our virus of interest, hCMV.

  1. Moreover, lack of T-cell CD8 and CD4 involvement can contribute to viral-induced cancer such as Kaposhi’s sarcoma (which caused Kaposi's Sarcoma-associated Herpesvirus, also a latent virus from the Orthoherpesviridae such as EBV and CMV), so a reference about T-cell role in oncogenesis would be a suggestion from my part.

Thank you for your observation. We have restructured this section and included the suggested information.

  1. Line 149: The new nomenclature is “Orthoherpesviridae”, not Herpesviridae.

Thank you for your observation; the nomenclature has been updated.

  1. It would seem that the acronym “hCMV” is appearing in brackets on multiple occasions within the text. It needs to appear only once, at the beginning (Lines 61,196, 202…).

Thank you for your observation. The full name Cytomegalovirus (hCMV) has been removed, and only the acronym hCMV is used throughout the manuscript.

  1. Although it is referenced by three papers, the phrase frol Lines 272-275 is not clear: “Viral surveillance and cancerous cells spreading are interplay mechanisms between tumours and viruses, this leads to an optimal microenvironment for viruses to replicate and at the same time increases malignant cells not to be targeted and destroyed by immune cells”. Can the Authors kindly elaborate what this interplay means? Can the Authors explain in more clear terms what “increases malignant cells” means?.

This paragraph has been rewritten and expanded for better clarity.

  1. Since, as the Authors write, “human IL10 and hCMV vIL10 displays a similarity of 40% to 415 50% across the different reference strains and isoforms”, could it be that only some stains of CMV are potentially cancerogenous, and others are not? I kindly suggest that the Authors add this observation to their work, since high-risk (HR) HCMV strains have been described in literature (Herbein, G. High-Risk Oncogenic Human Cytomegalovirus. Viruses 202214, 2462).

Thank you for your observation; it has been included based on the reference you provided.

  1. In the Conclusion section, the Authors offer a general insight into the thematic; however, I kindly suggest that the paragraph includes a sentence elaborating on the primary evidence supporting (or not supporting) the role of vIL-10. What does the reader keep from the text – is the literature more for or against implicating the vIL-10 in potential leukemogenesis?

Thank you for your suggestion; part of the conclusion was rewritten based on your input.

Round 2

Reviewer 1 Report

Comments and Suggestions for Authors

The responses to reviews were quite good, and the manuscript is improved. Some minor professional editing may be helpful.  

Reviewer 2 Report

Comments and Suggestions for Authors

Dear Authors,

Thank you for including my suggestions into your work. There is only one thing left, however. In the Conclusion section of the Manuscript, marked in yellow, there is an incomplete and typographically incorrect sentence (lines 646-648) :

"Continued research is essential to clarify the mechanisms that regulate hCMV latency and reactivation, as well as the proteins involved in these processes as well as different molecules part of the cellular fuctions which can be . "

The word "functions" is misspelled. Also, the sentence end would seem to lack a conclusion?

Please make sure that there are no other grammar errors or writing omission such as this throughout the text.

If this is corrected, I suggest the paper to be published.